# On the Safe Deployment of Matrix Multiplication in Massively Parallel Safety-Related Systems

**Javier Fernández** [1,2,*], **Jon Perez-Cerrolaza** [1], **Irune Agirre** [1], **Alejandro J. Calderon** [1], **Jaume Abella** [3] **and Francisco J. Cazorla** [3]

1   Ikerlan Technological Research Center, Basque Research and Technology Alliance (BRTA), 20500 Mondragon, Spain; jmperez@ikerlan.es (J.P.-C.); iagirre@ikerlan.es (I.A.); ajcalderon@ikerlan.es (A.J.C.)
2   Departament d'Arquitectura de Computadors, Universitat Politècnica de Catalunya (UPC), 08034 Barcelona, Spain
3   Barcelona Supercomputing Center, 08034 Barcelona, Spain; jaume.abella@bsc.es (J.A.); francisco.cazorla@bsc.es (F.J.C.)
*   Correspondence: javier.fernandez@ikerlan.es; Tel.: +34-616-597-463

**Abstract:** Deep learning technology has enabled the development of increasingly complex safety-related autonomous systems using high-performance computers, such as graphics processing units (GPUs), which provide the required high computing performance for the execution of parallel computing algorithms, such as matrix–matrix multiplications (a central computing element of deep learning software libraries). However, the safety certification of parallel computing software algorithms and GPU-based safety-related systems is a challenge to be addressed. For example, achieving the required fault-tolerance and diagnostic coverage for random hardware errors. This paper contributes with a safe matrix–matrix multiplication software implementation for GPUs with random hardware error-detection capabilities (permanent, transient) that can be used with different architectural patterns for fault-tolerance, and which serves as a foundation for the implementation of safe deep learning libraries for GPUs. The proposed contribution is complementary and can be combined with other techniques, such as algorithm-based fault tolerance. In particular, (i) we provide the high-performance matrix multiplication CUTLASS library with a catalog of diagnostic mechanisms to detect random hardware errors down to the arithmetic operation level; and (ii) we measure the performance impact incurred by the adoption of these mechanisms and their achievable diagnostic coverage with a set of representative matrix dimensions. To that end, we implement these algebraic operations, targeting CUDA cores with single instructions and multiple-thread math instructions in an NVIDIA Xavier NX GPU.

**Keywords:** safety; reliability; CNN; matrix multiplication; GPU; fault detection

## 1. Introduction

deep neural networks (DNNs) and convolutional neural networks (CNNs) are extensively used in autonomous systems to implement complex functionalities, such as object detection [1] and image classification [2]. CNNs comprise a large number of neurons distributed and interconnected in different layers. These structures have an intrinsic high level of computational parallelism that requires handling massive volumes of data. As a result, these applications demand high-performance computational capabilities, such as those provided by GPUs. The high level of computing parallelism of GPUs, together with their complex memory hierarchy, may spread a single random hardware error to multiple errors [3], jeopardizing the computation correctness. Furthermore, because the use of these platforms in the safety-critical domain is still in its infancy, there are no established common practice solutions to diagnose all internal components that take part in the computations, which are often barely documented by the platform manufacturer.

Besides, Figure 1 illustrates the possible consequences of a single error on a CNN-based object detector. We classify an image ("9e75b2a9-98437b5b.jpg" from "10K Image" package) from Berkeley DeepDrive dataset [4] with YOLOv3 [5] based on a pre-trained model with the COCO dataset ("yolov3.cfg" and "coco.data" extracted from the configuration folder (https://github.com/AlexeyAB/darknet/tree/darknet_yolo_v3, accessed on 12 March 2022). Activation weights from https://pjreddie.com/media/files/yolov3.weights, accessed on 12 March 2022) [6]. In the absence of errors, the CNN correctly detects 35 classes, such as people or traffic lights, among others (Figure 1a). However, a single-bit error injection in a weight of the first layer of the CNN leads to detecting 11 classes not present in the image, such as three wine glasses or four dining tables (Figure 1b), but no people or traffic lights. In contrast, if the error is injected at bit 705, it does not impact the CNN inference, becoming a latent error if further diagnostics are not in place. These errors are of particular concern in safety-related systems, where misclassifications can have catastrophic consequences.

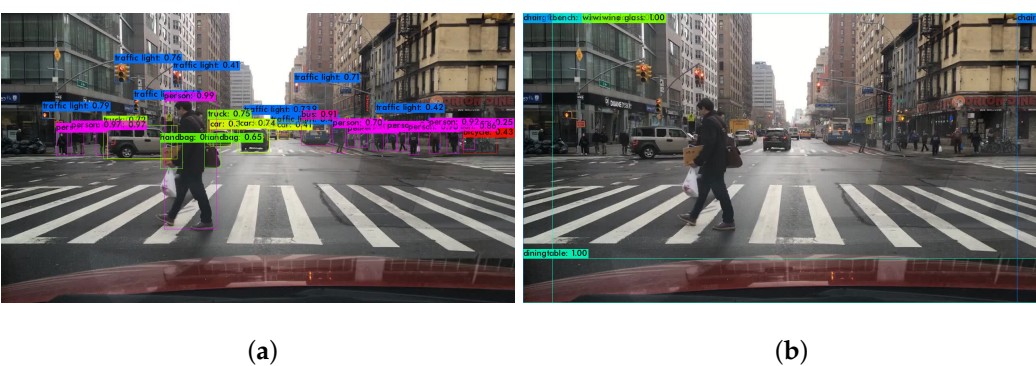

(**a**)                                          (**b**)

**Figure 1.** Example of a single-bit error impact in an object detection application based on CNNs. (**a**) Error-free or single-bit error (bit 705) inference; (**b**) Single-bit error (bit 734) inference.

Current solutions focus on selectively hardening the most vulnerable parts [7] or guaranteeing consistency among inputs and outputs matrices [8,9], but not on arithmetic operations or all combinatorial (e.g., muxes, bypasses) and sequential elements (e.g., latches) between memories and arithmetic units. In the context of commercial off-the-shelf GPUs, implementation-specific tests [10] or software-based self tests (SBSTs) [11] of individual components are not viable since they require fine-grain knowledge of the GPU component design and implementation. However, safety standards require not only guaranteeing that outputs are correct but also that internal components are periodically diagnosed to detect random hardware errors and reduce the probability of latent errors. Hence, existing built-in hardware safety mechanisms shall be complemented with software-based diagnosis mechanisms whenever the required diagnostic coverage is not achieved.

This paper paves the way towards the safe implementation of CNN-based safety solutions on massively parallel GPU-based platforms through the application of error-detection mechanisms using execution signatures (ESs), which also support an indirect functional diagnosis of used GPU components (without the need of component design and implementation details knowledge). To this end, we focus on the most computationally expensive operation of the CNNs, the matrix–matrix multiplication (MMM), that is the central part of machine learning libraries such as YOLOv3, accounting for 67% of YOLO's execution time [12]. In this way, and focusing on the CUTLASS library [13], we make the following contributions:

- We define and implement a catalog of diagnostic mechanisms to compute an array of ESs in a matrix–matrix multiplication defined in CUTLASS for GPUs. Additionally, we define two commonly used architectural safety patterns where this catalog can be implemented to support fault-tolerance;
- We analyze the performance impact and the diagnostic coverage (DC) of this catalog of mechanisms for multiple matrix dimensions in a NVIDIA Jetson Xavier NX GPU and

compare our results against sequential and advanced vector extensions (AVX)-based implementations [14];

- We perform an in-depth analysis of the tradeoff between DC and performance for a specific matrix dimension evaluating the required achievable DC by the IEC 61508 standard for different safety integrity levels (SILs) and architectural patterns.

The rest of the paper is structured as follows: Section 2 presents the required background. Section 3 describes the error-detection adaptations for the matrix–matrix multiplication. Section 4 describes and analyzes the results and makes a comprehensive analysis. Section 5 discusses related work. Finally, Section 6 concludes the paper and presents directions for future research.

## 2. Background

This section provides a brief introduction to the two main axes of our work: safety certification and CUTLASS. Additionally, acronyms are summarized in Table 1:

**Table 1.** Lists of acronyms.

| | |
|---|---|
| ABFT | algorithm-based fault tolerance |
| ABED | algorithm-based error detection |
| AVX | advanced vector extensions |
| CNN | convolutional neural network |
| DC | diagnostic coverage |
| DNN | deep neural network |
| CRC | cyclic redundancy check |
| DTI | diagnostic time interval |
| ES | execution signature |
| GPU | graphics processing unit |
| HFT | hardware fault tolerance |
| HPEC | high-performance embedded computing |
| MMA | matrix multiply–accumulate |
| MMM | matrix–matrix multiplication |
| PTX | parallel thread execution |
| SBST | software-based self test |
| SIL | safety integrity level |
| SIMT | single-instruction multiple thread |

### 2.1. Safety Certification

Safety-related systems are subject to certification, which is often achieved based on functional safety standards. IEC 61508 [15] is the reference standard for many domain-specific standards, such as road-vehicles [16] and railway [17]. These standards define the necessary requirements, techniques, and measures to avoid, mitigate, and detect random hardware errors and systematic errors. The SIL defines a safety integrity range where four is the highest and one is the lowest level (SIL1…SIL4). According to this safety criticality level, functional safety standards require the adoption of demanding safety measures and techniques. Particularly, standards require that systematic faults are duly mitigated through the measures adopted in the development process and that random hardware errors (transient and permanent) are detected, mitigated, and controlled at

runtime. In this context, ISO 26262 also refers to latent errors to denote errors not detected by safety mechanisms.

The assessment of the effectiveness of diagnostic mechanisms is generally evaluated in the form of DC. As defined in reference [18], *"Diagnostic Coverage (DC) denotes the effectiveness of diagnosis techniques to detect dangerous errors, expressed in coverage percentage with respect to all possible dangerous errors"*. IEC 61508 [15] classifies DC as low ($60\% < DC < 90\%$), medium ($90\% \leq DC < 99\%$), and high ($99\% \leq DC$). The required DC is determined by the SIL and hardware fault tolerance (HFT) (IEC 61508-2 Table 3). For example, if the architecture is not redundant (HFT = 0), a SIL1 safety function requires a low DC and a SIL3 requires a high DC. In a dual-channel redundant architecture instead (HFT = 1), a SIL3 safety function can be implemented with a medium DC. As stated in reference [18], the implementation of DC techniques based on software becomes relevant in order to diagnose the proper operation of the hardware components periodically (e.g., against permanent errors), or for the safe operation of the device concerning possible errors not detected by hardware built-in diagnosis (usually ranked as low or medium DC) or to complement them.

### 2.2. CUTLASS

CUTLASS is a collection of templates coded in CUDA and C++ that abstract the high-performance matrix-multiplication implementation. This open-source and low-level library decomposes this algebraic operation into software modules using C++ template classes. These modules divide matrix multiplication into threads, warps, blocks, and device levels, as can be seen in Figure 2. This figure highlights the memory transfer in each iteration of the most external loop of the matrix multiply–accumulate ($C \mathrel{+}= A \times B$). Additionally, CUTLASS allows tuning through custom data types, tiling sizes, and other algorithmic policies.

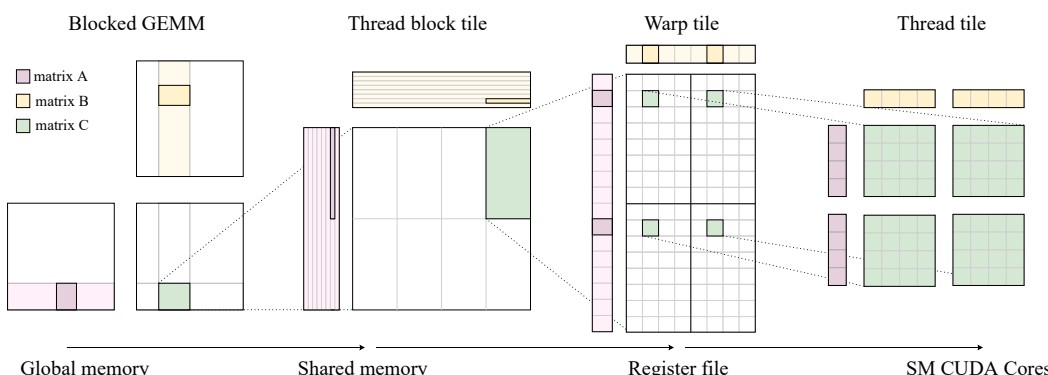

**Figure 2.** Cutlass GEMM hierarchy [13].

## 3. Enhancing MMM Safety

In this section, we elaborate a catalog of diagnostic techniques that compute an array of execution signatures to detect matrix–matrix multiplication execution errors. We also study their reproducibility and memory usage to minimize the performance impact. Finally, we also define two examples of safe architectural patterns in which the presented diagnostic catalog can be employed to support fault-tolerance.

### 3.1. Diagnostic Techniques

The main idea is to compute an array of ESs at runtime to enhance the matrix–matrix multiplication execution with error-detection capabilities. We propose to compute this array of ESs and protect the data employed at the arithmetic operation level by including a diagnostic mechanism or a combination of diagnostic mechanisms. These diagnostic techniques can be integrated using different safety architectural patterns, as will be explained in Section 3.4.

A set of existing checksum techniques provide varying degrees of DC and performance impacts [19,20]. Their usage has been focused on assuring integrity in data transmission [21], and as a result, their error-detection effectiveness has been widely studied. We refer the interested readers to references [19,20] for an in-depth discussion on checksums effectiveness from which we select the following: XOR, one's and two's complement, Fletcher, and cyclic redundancy check (CRC). Furthermore, we adapt these checksums to detect the execution errors of the matrix–matrix multiplication function extracted from CUTLASS [13] based on the single-instruction multiple-thread paradigm. At thread level, the CUTLASS library performs a matrix-multiply–accumulate operation in a triple-nested loop denoted as inner (I), intermediate (M), and external (E), as shown in Algorithm 1.

---

**Algorithm 1** MMM loops computed by a thread

---

```
 1: for each column of a tile from A matrix do
 2:       External loop statements
 3:       for each column of a tile from B matrix do
 4:             Intermediate loop statements
 5:             for each row of a tile from A matrix do
 6:                   Internal loop statements (compute the multiplication)
 7:                   [Checksum (I)]
 8:             end for
 9:             [Checksum (M)]
10:       end for
11:       [Checksum (E)]
12: end for
```

---

Depending on the employed checksum and the matrix–matrix multiplication loop at which it is applied, we obtain a wide catalog of solutions with varying levels of DC and performance impact. We classify them into two categories: (i) individual—in which we implement a single checksum in each matrix–matrix multiplication loop, and (ii) combinations—in which we implement combinations of two different checksums in the inner and intermediate loops. We design this latter category to offer additional trade-offs between DC and performance impact. In this way, we give the option to the safety engineer to choose from the different solutions of the catalog based on the specific DC and performance requirements of each safety application.

The GPU implementation of the checksums is performed as follows. XOR and one's and two's complement checksums and employ a low-level parallel thread execution (PTX) [22]. We code these checksums with an instruction set provided by this intermediate language, aiming to be portable across multiple GPU architectures. Among the benefits, the extended-precision integer arithmetic instructions allow holding carry-in and carry-out with a carry-bit flag in an integer-addition operation. This feature is highly desirable in one's complement checksum as it reduces the performance impact incurred by its inclusion. In Algorithm 2, readers can observe the one's complement checksum implementation based on PTX instructions:

---

**Algorithm 2** One's complement checksum

---

```
 1: function __A1C(uint32_t ui32_a, uint32_t ui32_b)
 2:       uint32_t acc;
 3:       asm ("add.cc.u32   %0, %1, %2; "
 4:             "addc.u32     %0, %0, 0; "
 5:             "not.b32      %0, %0; "
 6:             "=r" (acc)
 7:             "=r" (ui32_a), "r"(ui32_b));
 8:       return acc;
 9: end function
```

---

For the Fletcher, we employ the implementations provided by reference [14]. Finally, we resort to lookup tables for implementing the CRC. This method accelerates the protection of multiply–accumulate operations as follows: (1) we precompute the ES applying the CRC algorithm to all chunk's possible values of a prefixed number of bits (*n*). These values are stored in a $2^n$ lookup table; (2) we access these CRC values at runtime.

### 3.2. Reproducibility

Another crucial factor for the safe deployment of CNN algorithms in high-performance embedded computing (HPEC) platforms exploiting parallelism is the execution order. The matrix–matrix multiplication involves the use of floating-point data types that do not satisfy the associative property. Therefore, their use is considered as a source of numerical reproducibility errors [23,24]. In reference [25], NVIDIA highlights that despite all individual operations accomplished with the IEEE 754 standard [26], the result may not be bit-identical. Consequently, neither the order-independent checksums based on sums operations (XOR, one's and two's complement) nor the order dependent ones (Fletcher and CRC) can be implemented directly without assuring a deterministic execution order.

As a solution, we propose employing as many ESs as threads are involved in the matrix–matrix multiplication. Each thread executes a tile of the matrix–matrix multiplication in sequential order in a CUDA core. We use the global identifier of each thread to store the final ES computed by each thread at its relative address in an array of ESs. Note that, instead, combining ESs from different threads into a single ES would challenge reproducibility if we cannot enforce a specific computation order which is not trivial.

### 3.3. Memory Hierarchy

Applying previous checksum techniques to the matrix–matrix multiplication involves as many accesses to memory as the variables we intend to protect. As some implementations protect all variables involved in the matrix–matrix multiplication, the matrix–matrix multiplication computation entails protecting three times the number of performed matrix multiply–accumulate operations. In GPU-based implementations, the memory access speed is essential for achieving a good performance, mainly for real-time applications that are subject to strict execution-time requirements. Therefore, the type of memory chosen in the hierarchical memory model can be crucial. In our implementation, we allocate the ES into the global memory device. During the matrix–matrix multiplication execution, these ESs are transferred to registers (the memory with the highest speed) and returned to global memory when the computation is finished. In Figure 3, we depict the employed memory hierarchy. The grayscale used in the GPU memories denotes the memory access time, with the darkest gray requiring a higher latency.

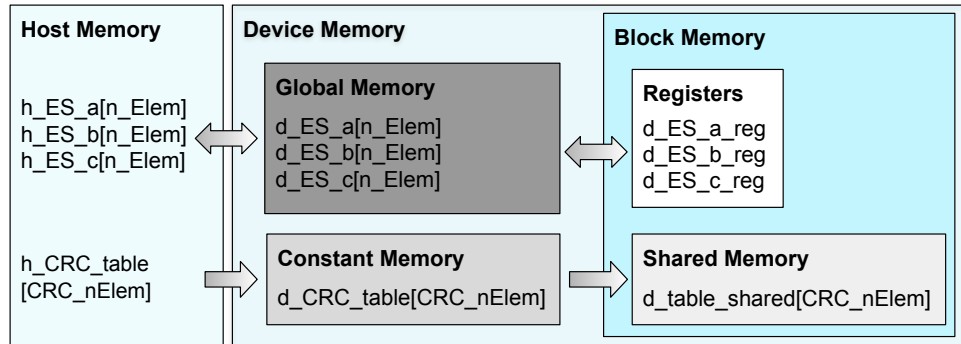

**Figure 3.** ES transference among GPU memory hierarchies (*CRC_nElem*: number of CRC lookup table memory addresses. *n_Elem*: number of ES memory addresses).

A particular concern arises in the CRC implementation, which requires further accesses to the lookup table. As all threads share these values, we have chosen shared memory instead of global memory to reduce memory access time. However, the shared memory has a reduced memory size that limits the CRC lookup table dimension. The safety designer

has to consider such a limitation at the design phase according to the target GPU's memory. In this paper, we perform CRC execution byte-by-byte, which requires $2^8$ shared memory addresses and, hence, four accesses to shared memory per protected 32-bit data word.

In Figure 3, we schematize the transfer of the CRC lookup table and the ESs across the memory hierarchy of the chosen platform. This transfer starts from the host memory, where it is initialized, to the shared memory, where it is finally accessed by each thread. Since GPU shared memory cannot be statically initialized, we initially store the CRC lookup table in constant memory before it is transferred to the shared memory at runtime. This memory presents a lower latency than global memory, especially in memory accesses where several threads access the same memory addresses consecutively. We can exploit this since the matrix–matrix multiplication executes as many blocks as $k$ tiles it is decomposed in, and all these blocks store the CRC table in their own shared memory. Although the first block accesses occur serialized, the remaining blocks access the constant cache memory (which has the required 1 KB to store the entire table) that is faster than the global memory.

### 3.4. Safety Architectural Patterns

Considering the safety measures proposed by current functional safety standards (ISO 26262, IEC 61508) and the architectural patterns described in reference [18] for HPEC platforms, we define two examples of safe architectural patterns that support the integration of the previously defined 'safe MMM' (see Figure 4) with different HFT levels (e.g., HFT = 0,1,2).

- *Periodic diagnosis with design-time fixed data pattern(s):* The 'safe MMM' is executed at least once every diagnostic time interval (DTI) (Figure 4b) with predefined design-time reference input data vectors that should lead to known reference outputs and an array of ESs values (see Figure 4a). This approach can be used in a single-channel architecture (HFT = 0) or in redundant architectures (e.g., triplicated architecture with HFT = 2). The error detection can be used to detect the erroneous channel prior to the voting process and application-specific measures can be implemented (e.g., restart erroneous channel, activate safe state);

- *Redundancy (with or without diversity):* This pattern redundantly executes the 'safe matrix–matrix multiplication' with $n$ replicas, each of which uses the same real-time input data and generates both an output and ESs array (Figure 4c) for each computation cycle (Figure 4d). A voting mechanism continuously compares those ESs arrays (and even the outputs), discarding the replicas with discrepancies and/or implementing application-specific measures (e.g., restart erroneous channel).

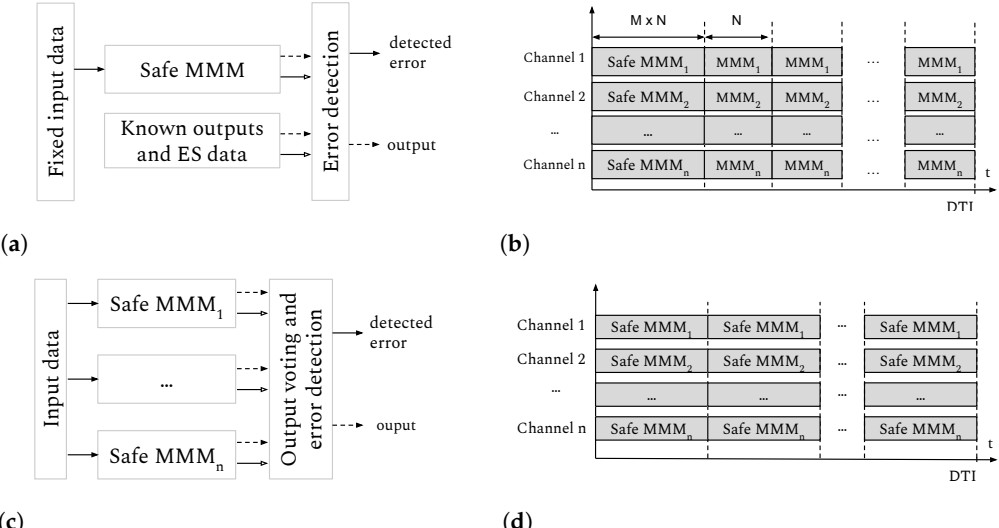

**Figure 4.** Safety architectural patterns. (**a**) Periodic diagnosis pattern; (**b**) Periodic diagnosis pattern scheduling; (**c**) Redundancy pattern; (**d**) Redundancy pattern scheduling.

## 4. Evaluation

Next, we evaluate the DC and performance impact incurred by the different diagnostic techniques in the matrix–matrix multiplication.

### 4.1. Experimental Set-Up

We use an NVIDIA Jetson Xavier NX platform, designed to accommodate DNN applications. We use the *clang* compiler with CUDA (both version 10) in an Ubuntu system. In order to minimize system interference, we employ the PREEMPT-RT patch and isolate the NVIDIA Carmel ARM core that executes the program with the highest real-time priority and configure it to run at the maximum frequency. We launch a single matrix–matrix multiplication stream to the GPU to avoid the uncertainty in the order of execution of the streams associated with several applications running simultaneously [27]. We employ the same matrix dimensions and nomenclature as in reference [14], seeking a fair comparison. In reference [14], the authors also include a catalog of diagnostic techniques, but based on a sequential and an AVX-based implementation. Table 2 shows the matrix–matrix multiplication dimensions, dividing them into two groups:

- Square matrices: seek to evaluate the influence of matrix size. We represent the dimensions of input matrices *A* and *B* and output matrix *C* as $N \times N$;
- Unbalanced matrices: focus on assessing the representativeness of the matrix dimensions in performance impact experiments and assessing the variability of the DC when modifying the relationship between rows and columns. We denote the matrix dimensions of *A* as $M \times K$, *B* as $K \times N$, and *C* as $M \times N$. In the performance results, *L*91 refers to a particular layer position extracted from a CNN. We follow the same notation in DC.

**Table 2.** Matrix dimensions employed in the experiments.

| Experiments | Square | | Unbalanced | | | |
| --- | --- | --- | --- | --- | --- | --- |
| | N | N | M | N | K | Name |
| Performance Impact | 80 | 80 | 18 | 230,400 | 64 | L91 |
| | 160 | 160 | | | | |
| | 320 | 320 | | | | |
| Diagnostic Coverage | 20 | 20 | 32 | 29 | 144 | L1 |
| | 40 | 40 | 8 | 900 | 8 | L2 |
| | 80 | 80 | 15 | 225 | 48 | L3 |

In performance impact experiments, we run the matrix–matrix multiplication function 1000 times and measure the execution time with <time.h> library to reach a nanoseconds resolution and calculate the mean value. We disregard the initial 100 measurements to avoid the cold-start problems associated with caches and the delays associated with the initial kernel launches [28].

In DC experiments, we perform a bit-exhaustive fault-injection campaign in both *A* and *B* matrices (as in reference [14]). This evaluation requires as many matrix–matrix multiplication executions as bits positions have those matrices. This is the reason for employing smaller matrix dimensions in DC in contrast with performance impact experiments. We have performed these bit-flips in the host before launching the kernels executing the matrix–matrix multiplication in the GPU.

### 4.2. Performance Impact

In order to derive the relative performance impact, we divide the execution time of the matrix–matrix multiplication, including the diagnostic techniques, by the execution time of the original matrix–matrix multiplication, with identical matrix dimensions and compiler optimization. This performance impact is relative to single-matrices execution.

However, performance impact varies at system level depending on the safety architectural pattern in which the safety designer includes the diagnostic mechanism(s). Thus, the impact on the periodic diagnostic pattern described in Section 3.4 is calculated by dividing the single 'safe matrix–matrix multiplication' execution by the DTI, not depending only on the performance impact of a single 'safe matrix–matrix multiplication' execution but also on the number of iterations of the matrix–matrix multiplication in the DTI. According to the implementation of redundant patterns, the performance impact can either: (i) be multiplied by a factor of two (in cases where the matrix–matrix multiplication is re-executed on the same hardware) or (ii) maintain the same value as that observed for single MMMs (double hardware cost). This subsection provides the performance impact relative to single-matrices execution. From these results and according to the implemented safety architectural pattern, safety designers can compute the final performance impact in their systems.

Initially, we perform timing experiments by disabling compiler optimizations to avoid additional safety challenges brought by optimizations (compiler option -O0). Throughout this paper, we use compiler optimizations to refer to both host and device compilers. In this way, we perform the performance impact experiments with the same optimizations for both. Figure 5 shows how the performance impact caused by the inclusion of both individual or a combination of checksums is rather small (between 1× and 1.25× for all matrix dimensions). In reference [14], the authors show that increasing matrix dimension sizes lead to decreasing performance impacts for sequential and AVX-based implementations. Furthermore, we also show significant differences across different checksums. Moreover, in Figure 5 we observe a very similar impact for different matrix dimensions in our GPU-based implementation, as well as across different checksums. This behavior relates to the fact that execution time is dominated by memory accesses. Hence, arithmetic operations due to the different checksums have a very limited impact that barely changes when varying the matrix dimensions or the specific checksum used.

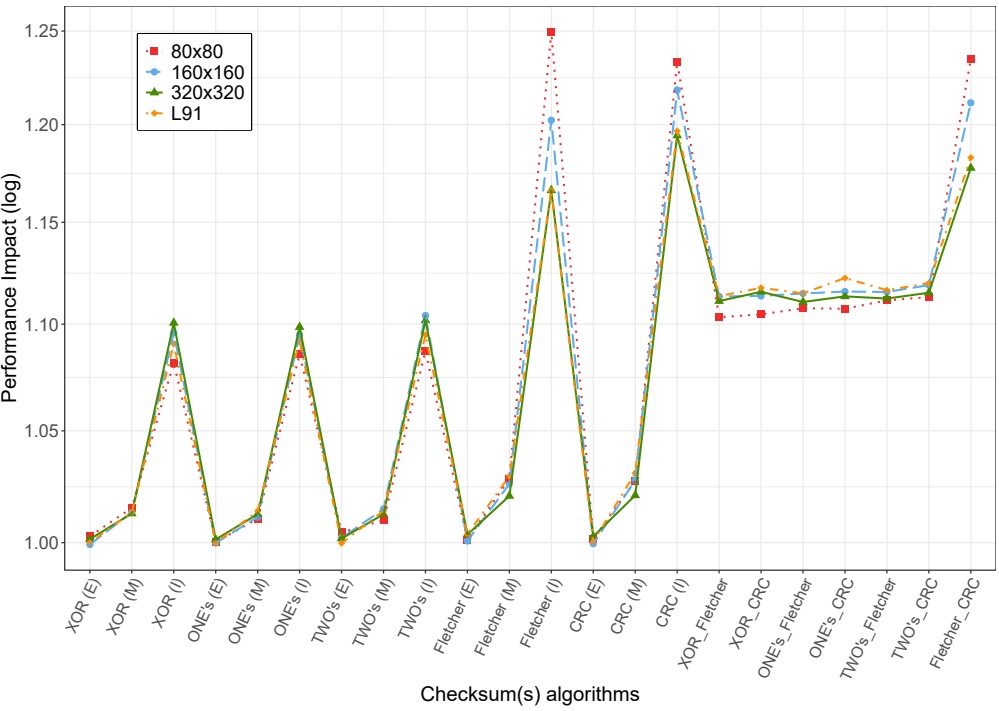

**Figure 5.** Performance impact with -O0 optimization.

Since real-time safety applications may need optimizations to achieve the required performance, we evaluate the performance impact with a higher compiler optimization too (compiler optimization option -O3). In Figure 6, we can observe that the performance impact increases across all checksums, but particularly for the Fletcher and CRC if imple-

mented in the internal loop. Such impact further exacerbates when increasing the matrix dimensions, as opposed to the case of sequential and AVX-based implementations. CRC implementation involves four accesses to the shared memory to protect each word in the loop. In GPUs, memory latency is crucial from a performance point of view. That explains the performance impact produced by the CRC implementation, reaching an impact of up to $\approx 100\times$ the original implementation. Concerning the Fletcher implementation, the performance impact relates to the modulo operation, used twice in each protected word, whose implementation is not efficient in NVIDIA GPUs compared to the sequential implementation evaluated in reference [14]. These results show that with optimizations not all checksum combinations may be affordable in terms of computing performance impact in contrast with -O0. It should be noted that with -O3 the MMM execution time is three orders of magnitude smaller than with -O0. This explains the higher relative impact of -O3 optimization experiments.

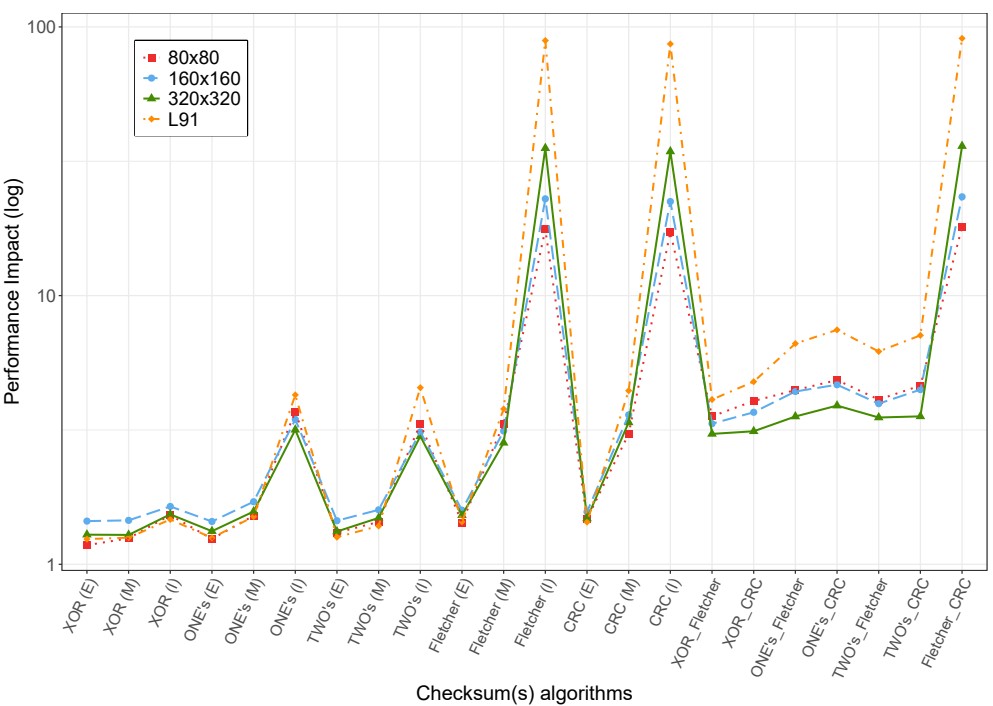

**Figure 6.** Performance impact with -O3 optimization.

### 4.3. Diagnostic Coverage

We have performed a fault-injection campaign (following the same methods used in reference [14]) in input matrices *A* and *B* in all bit positions before launching the execution in the GPU to evaluate the achievable DC with every single-diagnostic mechanism implementation and the combinations included in our catalog. Initially, we perform the matrix–matrix multiplication without fault-injections, and we store the ES computed as the reference or golden value. Then, we perform the single-bit fault-injection campaign and the ESs comparison with the golden value for each execution. Finally, we quantify the DC as the number of detected faults relative to the total number of injected faults.

Table 3 gathers the results of our fault-injection campaign implemented in GPU together with the DC of the sequential and AVX-based implementations extracted from reference [14], excluding the combinations that reach 100% in all cases. This table shows that GPU-based implementations achieve a higher DC than the less parallelized implementations for external (E) implementations. This occurs due to the specific implementation details of the MMM for the GPU, where the entire matrix is decomposed into block tiles that independently compute partial matrix multiplications. In this implementation, the number

of values protected in the most external loop increases with a consequent increment in the achievable DC.

We observe the same trend in the intermediate implementations in square matrices with dimensions greater than 20 and unbalanced matrices L2 and L3, with an exception in the XOR implementation in L3, which was motivated by the nature of XOR which does not detect even single-bit errors. In the rest of the matrices, the DC of AVX is superior to that achieved in GPUs. The reason lies in the AVX-based implementation that protects eight values in the intermediate loop and, in small matrices, the number of values protected is larger than those for the GPU.

Internal loop implementations reach 100% DC in all matrix dimensions with one's complement, Fletcher, and CRC. XOR implementation has identical DC to sequential and AVX in all dimensions except in the L3 matrix. DC drops because our GPU-based implementation for these matrix sizes divides the matrix–matrix multiplication into an even number of computations performed by each thread, and this checksum fails to detect even failures. Two's complement provides similar results to the other implementations.

Finally, all combinations of checksums reach the maximum DC, excluding the ones presented in Table 3 that still provide a high DC. In the GPU-based implementation, the combinations, including Fletcher in the intermediate loop, do not provide as much DC as the other implementations. This relates to the location of the Fletcher checksum computation in the code, which could not be kept identical to the other implementations in the GPU-based implementation.

**Table 3.** DC in sequential [14], AVX-based [14], and GPU-based implementations.

| Checksum Implemented | Square | | | | | | | | | Unbalanced | | | | | | | | |
| | 20 | | | 40 | | | 80 | | | L1 | | | L2 | | | L3 | | |
| | Seq | AVX | GPU | Seq | AVX | GPU | Seq | AVX | GPU | Seq | AVX | GPU | Seq | AVX | GPU | Seq | AVX | GPU |
|---|---|---|---|---|---|---|---|---|---|---|---|---|---|---|---|---|---|---|
| XOR (E) | 2.5 | 2.5 | 10.0 | 1.3 | 1.3 | 10.0 | 0.6 | 0.6 | 12.5 | 0.4 | 0.4 | 6.6 | 0.1 | 0.1 | 12.4 | 0.1 | 0.1 | 12.1 |
| XOR (M) | 50.0 | 50.0 | 50.0 | 50.0 | 50.0 | 50.0 | 50.0 | 50.0 | 50.0 | 52.5 | 52.5 | 52.5 | 0.9 | 0.9 | 0.9 | 6.6 | 10.0 | 6.3 |
| XOR (I) | 50.0 | 50.0 | 50.0 | 50.0 | 50.0 | 50.0 | 50.0 | 50.0 | 50.0 | 52.5 | 52.5 | 52.5 | 0.9 | 0.9 | 0.9 | 100.0 | 100.0 | 6.3 |
| One's (E) | 2.5 | 2.5 | 10.0 | 1.3 | 1.3 | 10.0 | 0.6 | 0.6 | 12.5 | 0.4 | 0.4 | 6.6 | 0.1 | 0.1 | 12.4 | 0.1 | 0.2 | 12.1 |
| One's (M) | 52.5 | 79.2 | 62.5 | 51.2 | 59.2 | 62.5 | 50.6 | 54.4 | 62.5 | 54.1 | 72.9 | 63.9 | 1.0 | 2.2 | 25.7 | 7.1 | 9.9 | 30.0 |
| One's (I) | 98.5 | 99.2 | 100.0 | 97.7 | 99.2 | 100.0 | 96.9 | 93.8 | 100.0 | 98.4 | 98.9 | 100.0 | 97.7 | 99.2 | 100.0 | 96.9 | 99.9 | 100.0 |
| Two's (E) | 2.5 | 2.5 | 10.0 | 1.3 | 1.3 | 10.0 | 0.6 | 0.6 | 12.5 | 0.4 | 0.4 | 6.6 | 0.1 | 0.1 | 12.4 | 1.0 | 0.2 | 12.1 |
| Two's (M) | 52.3 | 68.8 | 61.7 | 51.1 | 59.1 | 61.7 | 50.6 | 54.2 | 61.7 | 54.1 | 63.5 | 63.2 | 1.0 | 1.7 | 24.1 | 7.1 | 9.6 | 28.5 |
| Two's (I) | 96.9 | 96.9 | 95.7 | 95.3 | 95.3 | 95.7 | 93.8 | 99.2 | 95.7 | 98.4 | 92.6 | 95.9 | 90.7 | 90.7 | 91.5 | 96.9 | 100.0 | 92.0 |
| Fletcher (E) | 2.6 | 3.5 | 10.0 | 1.3 | 1.5 | 10.0 | 0.6 | 0.7 | 12.5 | 0.4 | 0.5 | 6.6 | 0.1 | 0.2 | 12.4 | 0.1 | 0.2 | 12.1 |
| Fletcher (M) | 52.2 | 68.8 | 62.5 | 51.1 | 60.0 | 62.5 | 50.6 | 55.0 | 62.5 | 54.1 | 73.8 | 63.9 | 1.0 | 2.2 | 25.7 | 7.1 | 10.0 | 30.0 |
| Fletcher (I) | 100.0 | 96.9 | 100.0 | 100.0 | 100.0 | 100.0 | 100.0 | 100.0 | 100.0 | 100.0 | 100.0 | 100.0 | 10.0 | 100.0 | 100.0 | 100.0 | 100.0 | 100.0 |
| CRC (E) | 2.6 | 3.5 | 10.0 | 1.3 | 1.5 | 10.0 | 0.6 | 0.7 | 12.5 | 0.4 | 0.5 | 6.6 | 0.1 | 0.2 | 12.4 | 0.1 | 0.2 | 12.1 |
| CRC (M) | 52.5 | 80.0 | 62.5 | 51.3 | 60.0 | 62.5 | 50.6 | 55.0 | 62.5 | 54.1 | 73.8 | 63.9 | 1.0 | 2.2 | 25.7 | 7.1 | 10.0 | 30.0 |
| CRC (I) | 100.0 | 100.0 | 100.0 | 100.0 | 100.0 | 100.0 | 100.0 | 100.0 | 100.0 | 100.0 | 100.0 | 100.0 | 100.0 | 100.0 | 100.0 | 100.0 | 100.0 | 100.0 |
| XOR_Fletcher | 99.8 | 100.0 | 90.6 | 99.8 | 100.0 | 90.6 | 99.8 | 100.0 | 90.6 | 99.8 | 100.0 | 91.0 | 99.8 | 100.0 | 81.4 | 99.8 | 100.0 | 82.5 |
| One's_Fletcher | 100.0 | 100.0 | 96.3 | 100.0 | 100.0 | 95.6 | 100.0 | 100.0 | 95.6 | 100.0 | 100.0 | 95.5 | 100.0 | 100.0 | 90.7 | 100.0 | 100.0 | 91.3 |
| Two's_Fletcher | 97.9 | 100.0 | 96.3 | 97.8 | 100.0 | 95.6 | 97.7 | 100.0 | 95.6 | 99.6 | 100.0 | 95.5 | 99.8 | 99.9 | 90.7 | 99.6 | 100.0 | 91.3 |

### 4.4. Trade-Off between Performance Impact and DC

In this subsection, we analyze the suitability of our catalog of diagnostics for an $80 \times 80$ square matrix–matrix multiplication, searching for a proper trade-off between DC and performance impact. As explained in a previous subsection, the performance impact varies substantially from the non-optimized compilation to the highest optimization, which has led to assessing both implementations, as can be seen in Figure 7.

Regardless of the compiler optimizations, both reach the same DC. However, in -O0 experiments, we obtain performance impacts between 1.01 and 1.23 in contrast with -O3, which increases this range to 1.18–17.98. We can confirm that combining one's and two's complement checksums with Fletcher is not suitable, reducing DC in contrast with the single-internal-loop implementation of these diagnostics.

We can realize from Figure 7 that all combinations, including CRC in the intermediate loop as well as the individual implementations of Fletcher, CRC, and one's complement in the most internal loops, allow us to reach 100% DC. Among all of them, the one's complement incurs the smallest performance impact and is, therefore, the most suitable

for detecting single-bit errors for this matrix–matrix multiplication dimension (1.09 in non-optimization compilation and 3.70 the highest compiler optimization). For medium DC, the most suitable option for -O0 compilation remains the same but with -O3 optimization the best trade-off can be reached with two's complement (I) (3.33 performance impact). Finally, the best performance for low DC is achieved in both optimizations with two's complement (M) (1.01 and 1.44 performance impact).

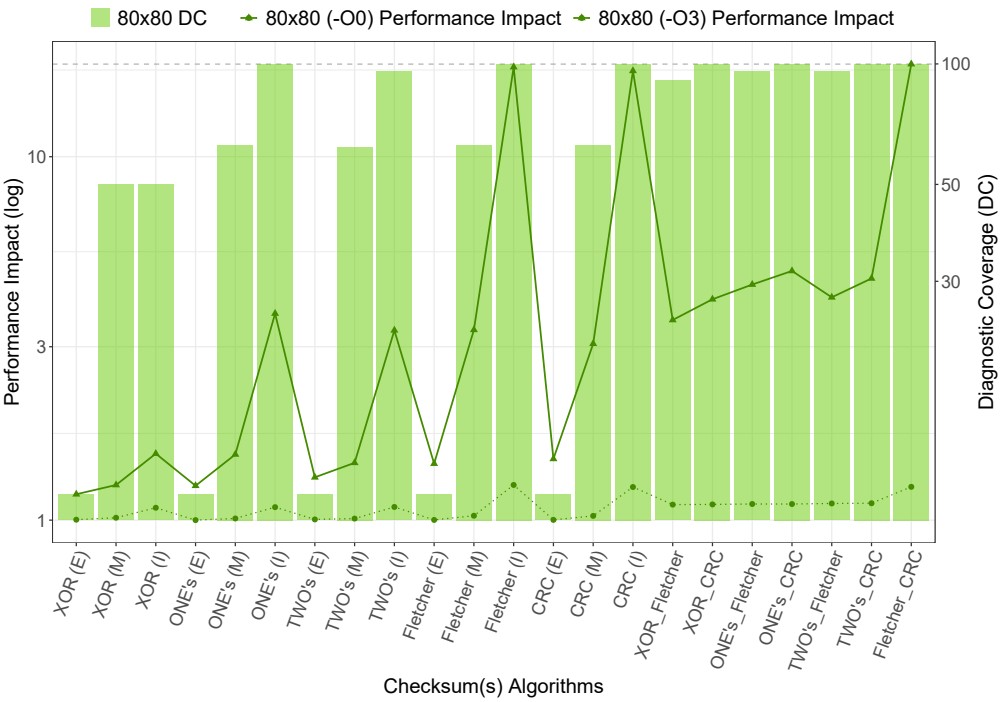

**Figure 7.** Trade-off between performance impact vs. DC.

After analyzing the most suitable checksums according to the DC range, performance impact, and compiler optimization for $80 \times 80$ square matrix dimensions, we examine the DC required to achieve the SIL and HFT established in the functional safety standards. For that, we recur to IEC 61508-2 in Table 2, where those relationships are defined. Notice that in practice, although the standards do not specify it, conscientious safety system designs for those HFT and SIL combinations involving DC below 60% (e.g., SIL = 1 with HFT = 1) require a DC closer to 60% than 0%. As DC and performance impact are MMM dimension-dependent—safety designs have to analyze both values based on the dimensions of their particular CNN. As a representative example, in Table 4 we provide the most suitable checksums and the incurred performance impact to reach the required DC for each HFT and SIL with square matrices with dimensions of 80x80. We employ a grayscale to denote the DC ranges defined in IEC 61508, with higher DCs being darker gray cells.

**Table 4.** Selected checksums for $80 \times 80$ matrix dimension according to SIL and HFT.

| | HFT | | | | | |
| | 0 | | 1 | | 2 | |
| **SIL** | **-O0** | **-O3** | **-O0** | **-O3** | **-O0** | **-O3** |
| 4 | Non achievable | | One's (I) (1.09)[iv] | One's (I) (3.7) [iv] | One's (I) (1.09)[iii] | Two's (I) (3.33)[iii] |
| 3 | One's (I) (1.09) [iv] | One's (I) (3.7) [iv] | One's (I) (1.09) [iii] | Two's (I) (3.33) [iii] | One's (M) (1.01) [ii] | Two's (M) (1.44) [ii] |
| 2 | One's (I) (1.09) [iii] | Two's (I) (3.33) [iii] | One's (M) (1.01) [ii] | Two's (M) (1.44) [ii] | XOR (M) (1.01) [i] | XOR (M) (1.25) [i] |
| 1 | One's (M) (1.01) [ii] | Two's (M) (1.44) [ii] | XOR (M) (1.01) [i] | XOR (M) (1.25) [i] | Non specified | |

NOTE: [i] $DC < 60\%$ [ii] $60\% < DC < 90\%$ [iii] $90\% \leq DC < 99\%$ [iv] $99\% \leq DC$.

In Table 4, the reader can observe the influence of the compiler optimization option in the election of the most appropriate diagnostic mechanism. For instance, for a SIL = 2 with a single-channel implementation (HFT = 0), the chosen checksum changes from one's (M) to two's (M) according to the compilation (-O0 and -O3, respectively).

## 5. Related Work

The functional safety certification challenges of DNN and CNN-based software solutions have been studied recently [9,29,30]. MMMs are at the heart of DNN and CNN software solutions; for example, over 90% of a CNN execution time is due to matrix–matrix multiplication-based convolutions [9] and matrix–matrix multiplication accounts for 67% of YOLO's execution time [12].

Several fault-injection experiments have analyzed the reliability of GPU-based CNN software implementations (e.g., references [31,32]). These analyses are fundamental to understanding the importance of random-hardware-errors management, and understanding the nature of errors and their propagation. Concerning GPU-based matrix–matrix multiplication software implementations, different technical approaches have been proposed for random-hardware-errors management (detection, correction, and mitigation), ranging from GPU and matrix–matrix multiplication software implementation-specific techniques to generalizable techniques.

GPU device and software implementation-specific techniques (e.g., references [7,33]) can potentially provide high error-detection rate claims with a low performance impact (e.g., detecting 84.5% of errors that lead to misclassification with 0.3035% performance impact [7]). However, the analysis, selection, evaluation (e.g., DC estimation), and implementation of techniques become software implementation- and GPU device-specific (reduced portability). This effort, and the impact analysis due to software or GPU device updates, is not neglectable in the context of safety-critical systems development and certification. Analogously, SBST [11] for specific components generally builds on low-level knowledge of the device under testing (e.g., gate-level implementation of the GPU) to devise software-only solutions with high coverage, and hence, SBST is neither portable across designs nor usable in COTS GPUs, whose low-level (circuit) design is unknown (no public information is available).

Generalizable techniques for matrix–matrix multiplication algorithms, such as algorithm-based error detection (ABED) and algorithm-based fault tolerance (ABFT) [9], are algorithm-specific dependent but less dependent on specific GPU architectures. ABFT leverages matrix–matrix multiplication algorithmic knowledge to provide fault-tolerance with the detection and correction of random hardware errors with performance impacts as low as 20% for square matrices or higher than 50% for non-square matrices [9]. Moreover, ABED-based techniques can potentially lead to high error-detection rate claims with low performance impact (e.g., 100% hardware errors with 6–23% performance impact) [9]. However, these techniques need to consider software implementation restrictions, such as considered data type (e.g., fixed-point integers or rounding errors with floating-point numbers [9]) and detect errors (only) in the generated output values. Moreover, since their coverage is much more limited than that of ESs, ABED-based techniques cannot be used for periodic diagnostics and need to be used continuously, which leads to high costs for real-time applications, such as autonomous driving, where ESs allow for a tailoring of the cost by setting the diagnostics frequency accordingly.

While these techniques focus on the reliability improvement of the matrix–matrix multiplication algorithm itself, the safety developer needs to consider several technical challenges not covered by previously described techniques. For example, the need to perform a periodic diagnosis of the GPU components used in such computations (e.g., memory, cache, warp scheduler, arithmetic logic unit) with the required DC during the periodic test interval in order to detect transient and permanent errors, and reduce the probability of undetected latent errors. The proposed generalizable safe matrix–matrix multiplication technique provides both an indirect diagnosis of the GPU components used in such computations

and the MMM execution itself, which require a bit-exact array of ESs considering also floating-point data types, and using different architectural patterns to achieve HFT levels (e.g., single-channel execution with HFT = 0, triple-redundancy execution with HFT = 2).

This technique should be considered complementary to the previously described techniques. For example, the safe matrix–matrix multiplication could be executed once every diagnostic test interval (with higher performance-impact cost), while the rest of the executions could be based on the original matrix–matrix multiplication algorithm or ABFT/ABED-based matrix–matrix multiplication algorithms that can further improve the overall reliability or error-detection coverage with a lower computational performance impact.

## 6. Conclusions and Future Work

This paper describes and evaluates the DC and performance impact of a catalog of diagnostic techniques integrated into the high-performance matrix–matrix multiplication CUTLASS library ('safe matrix–matrix multiplication') to detect GPU transient and permanent errors. This 'safe matrix–matrix multiplication' software implementation can be used to detect random hardware errors in the matrix–matrix multiplication software execution itself and perform an indirect diagnosis of the GPU components that compute the matrix–matrix multiplication, to detect permanent and transient errors, and reduce the probability of undetected latent errors. This paper paves the way toward the safe implementation of CNN-based safety solutions implemented in GPU-based platforms by enhancing the reliability of the most computationally expensive component of DNN in general and CNN in particular, the MMM, and represents a step forward in adherence to the current functional safety standards of safety systems involving ML components implemented in high-performance embedded computing platforms. With the selected GPU, the developed 'safe matrix–matrix multiplication' software implementation and $80 \times 80$ square matrix dimensions, low, medium, and high DCs are achieved with a minimum of 1.01, 1.09, and 1.09 performance impacts for the -O0 compiler optimization option and 1.44, 3.33, and 3.7 for the -O3 option.

Moreover, the 'safe matrix–matrix multiplication' can be integrated in different safety architectural patterns with different diagnostic approaches (e.g., design-time fixed pattern, real-time input data) and different fault-tolerance levels based on redundant architectures. Furthermore, as explained in the related work (Section 5), this approach should be also be considered potentially complimentary with respect to other existing techniques, such as ABFT and ABED.

The 'safe matrix–matrix multiplication' technique is generalizable to different GPU devices and matrix dimensions. However, achievable DC and the associated performance impact varies with GPU devices/architectures, software libraries, matrix dimensions, and compiler optimizations. Thus, whenever this technique is instantiated, experiments should be re-executed for the given GPU device/architecture, compiler, software library, and application-specific matrix dimensions to select the most suitable candidate from the catalog of integrated diagnostic techniques (e.g., DC vs. performance impact).

**Author Contributions:** Introduction and abstract, J.F., J.P.-C., I.A., A.J.C., J.A., and F.J.C. ; methodology, J.F., J.P.-C., I.A., A.J.C., J.A., and F.J.C.; software, J.F., J.P.-C., I.A., A.J.C., J.A., and F.J.C.; validation, J.F., J.P.-C., I.A., A.J.C., J.A., and F.J.C.; formal analysis, J.F., J.P.-C., I.A., A.J.C., J.A., and F.J.C.; investigation, J.F., J.P.-C., I.A., A.J.C., J.A., and F.J.C.; writing—original draft preparation, J.F., J.P.-C., I.A., A.J.C., J.A., and F.J.C.; writing—review and editing, J.F., J.P.-C., I.A., A.J.C., J.A., and F.J.C.; visualization, J.F., J.P.-C., I.A., A.J.C., J.A., and F.J.C.; supervision, I.A., J.P.-C., J.A., and F.J.C. All authors have read and agreed to the published version of the manuscript.

**Funding:** The research of this paper has received funding from the European Union's Horizon 2020 research and innovation programme (grant agreement No 871465 (UP2DATE)).

**Conflicts of Interest:** The authors declare no conflict of interest.

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
