# Peer review of "On the Safe Deployment of Matrix Multiplication in Massively Parallel Safety-Related Systems"

_applsci, doi:10.3390/app12083779_

Round 1
Reviewer 1 Report
The paper describes and evaluates the diagnostic coverage and performance impact of a series of diagnosis techniques tntegrated into the MMM CUTLASS llibrary to detec GPU transient and permanent errors.
The paper is technically sound. However, I have some concerns about it:
-
- The main concern is about the contribution of the paper on the field. It seems to be somewhat marginal, unless it is clearly estated in the Conclusions Section.
- DNN and other acronyms should be explained the first time they appear.
- Parentheses in some sentences throughout the Introduction should be removed since they don't coontribute nothing.
- The measure of performance impact in Figures 5 and 6 is not clearly explained. Is it the speedup?
Author Response
The authors would like to thank the reviewers for providing valuable comments to improve this paper. We have produced a new version addressing the concerns of the reviewers. All changes made in the paper are marked in blue color. Additionally, we number the reviewer’s concerns, associating them with an identifier that we include in the paper in orange boxes to clarify which one addresses each change. We attach a pdf with the response to the reviewers.

Reviewer 2 Report
There are some minor corrections in the text. Especially in the use of acronyms, be careful to define the meaning before the acronym.
It is very interesting how the behavior of the checksum algorithms varies depending on the optimization flag (-O0 and -O3). Apparently this flag affects executable code on the host, how does this flag affect operations performed on the GPU?
Author Response

(The authors gave the same response as above.)
